# An Overview of *Aquatica* Fu et al., a Phylogeny of Aquatic Fireflies Using Mitochondrial Genomes, a Description of Two New Species, and a New Record of Aquatic Fireflies in China (Coleoptera: Lampyridae: Luciolinae) [note 1]

**DOI:** 10.3390/insects15010031

**Published:** 2024-01-03

**Authors:** Xinhua Fu, Lesley Ballantyne

**Affiliations:** 1College of Plant Science and Technology, Huazhong Agricultural University, Wuhan 430070, China; fireflyfxh@mail.hzau.edu.cn; 2Hubei Insect Resources Utilization and Sustainable Pest Management Key Laboratory, Wuhan 430070, China; 3Firefly Conservation Research Centre, Wuhan 430070, China; 4School of Agricultural, Environmental and Veterinary Sciences, Charles Sturt University, P.O. Box 588, Wagga Wagga 2678, Australia

**Keywords:** firefly, aquatic, mitochondrial genome, phylogenetic analysis, morphology, female reproductive, integrative taxonomy

## Abstract

**Simple Summary:**

When we refer to aquatic fireflies, we are actually referring to their larvae, as the adults are free-flying. Many species of fireflies have been collected from terrestrial specimens only and may not have any information to indicate the larval habitat. The ideal situation is to rear specimens from eggs laid, but this is a difficult venture and is not always successful. More recently, the use of molecular genetics, especially the genome of the mitochondrion, has enabled us to match larvae with adults with great confidence. Here, we address two new species of aquatic fireflies from China where the aquatic association is confirmed, and the relationships of these species with other aquatic fireflies are established.

**Abstract:**

There are now seven species in the genus *Aquatica* Fu & Ballantyne, with all but one known from the characteristics of males, females, and larvae. Molecular information is combined with morphological taxonomy for the delimitation of both genus and species. The monophyly of the genus *Aquatica* is strongly supported in six trees, and its position as sister to the genus *Nipponoluciola* Ballantyne Kawashima Jusoh et Suzuki is stable across maximum likelihood and Bayesian inference results. Two new species of *Aquatica* Fu & Ballantyne, *A. qingshen* sp. nov. and *A. xianning* sp. nov., described from the features of males, females, and larvae, are closely related within a single clade within the genus *Aquatica*. Females have distinctively shaped median oviduct plates. A Chinese population identified as *A. lateralis* was found to be morphologically similar to the Japanese population, but genetic distances suggest that it is a distinctive species. No larvae are associated with this species. Definitions of the aquatic status of Luciolinae fireflies are expanded.

## 1. Introduction

Aquatic fireflies are well known in countries such as Japan and China, and they belong to the genera *Aquatica* Fu & Ballantyne, *Sclerotia* Ballantyne, and *Nipponoluciola* Ballantyne Kawashima Jusoh et al. Suzuki [1,2,3,4] within the subfamily Luciolinae. *Aquatica* was defined first by the morphological features of males, females, and larvae [1]. Molecular analyses addressing one or more of the species assigned to *Aquatica* by Fu et al. [1] investigated different aspects, including the luciferase gene [5,6,7]. The majority of analyses now address the complete mitogenome [4]. However, most address only one or two of the species, which are now placed in *Aquatica*. A well-defined *Aquatica* clade including at least four species is less often identified [8,9,10,11] but is not accompanied by morphology. Morphology also has shortcomings, but Ahrens [12] recommends that “morphology SHALL become a mandatory diagnostic trait in the ICZN and should obligatorily accompany any other diagnoses of a species being recognised by an alternative species concept”, which we endorse here.

While primarily male characteristics define the Luciolinae with aquatic larvae, larval and female taxonomy are also addressed. However, larval taxonomy in particular is still inadequate [3,13]. We can reliably assign the metapneustic back-swimming larvae to the genus *Sclerotia*, and soft-bodied bottom-feeding larvae with lateral abdominal gills belong to *Nipponoluciola* and *Aquatica*, but we are unable to differentiate individual species of *Sclerotia* larvae [4]. There are no obvious larval characters apart from colouration to differentiate the two genera with bottom-feeding larvae [4]. 

It is only within the Luciolinae that aquatic larvae (either with abdominal gills or with terminal spiracles) that spend their entire larval period in the aquatic environment are known [14]. Definitions of the aquatic status of beetles are critical in assessing the conservation of often fragile tracts of water, but they differ in the life cycle stage addressed. True water beetles were differentiated if their adults were submerged or partly so for most of their adult life, while false water beetles were aquatic only in their larval stage, with terrestrial adults [15]. Within the Luciolinae, given that all the adults observed were terrestrial and free-flying, a simplified classification system was adopted for larvae [3,4], with categories of aquatic, semiaquatic (equivalent to false and facultative water beetles, respectively, in [15]), and terrestrial. 

It is difficult to assess either the veracity or identification of some records of aquatic status (see the overview in [16]); Bertrand [17] (p. 599, Figures 1 and 2) confusingly keyed *Luciola* larvae with branched gills on the meso- and metathorax, and incorrectly represented *Luciola cruciata* Motschulsky larva (now *Nipponoluciola cruciata*) with meso- and metathoracic gills (gills are present only along the sides of the abdomen; see [4]). Bertrand [18] (Figures 8–10) recorded unidentified specimens of aquatic larvae from Sri Lanka with eight sets of abdominal gills and mandibles with an inner basal tooth, differentiated from Blair’s *Pyrophanes* Olivier larva and which may represent a *Pygoluciola* Wittmer species, which has not, until now, been recorded in Sri Lanka. Establishing the aquatic status, by rearing either eggs to larvae, or larvae to adults, still proves difficult. Of the fifteen reported cases of aquatic larval stages, the aquatic lifestyle could be confirmed in only five cases, all from China [16]. 

The aim of this study is to contribute to the knowledge of *Aquatica* by providing descriptions of two new species and one new record. Additionally, this study aims to update, consolidate, and reassess the morphological, taxonomic, and historical information available for the genus and the varying definitions of aquatic status. Notably, the investigation includes a novel examination of aspects of the female reproductive system. We also conduct phylogenetic analyses based on the mitogenome to determine the generic phylogenetic position and the relationships among its constituent species, as well as to assess its monophyly.

## 2. Materials and Methods

### 2.1. Morphology

Morphological characters of males, females, and larvae and interpretation of internal female anatomy follow [4] and larval morphology in [13], which defined larval types and expanded certain larval characters.

Unpublished female reproductive anatomy in [19] was confirmed and expanded here.

Procedures for dissection are outlined in [4] and references therein. We follow [3] for nomenclature relating to orientation of aedeagi in repose, referring to the side of the median lobe with the ejaculatory orifice as the ventral surface. The aedeagal sheath has clearly defined dorsal and ventral surfaces by comparison.

All illustrations depict the specimen either with the anterior end towards the top of the page or with the anterior end specified in the figure legend.

In the interests of brevity, most of the references from the extensive synonymic tables for *Aquatica lateralis* (Motschulsky) and other *Aquatica* species are not listed here and can be found in [1,2,3,4]. 

Ballantyne used an Olympus SC100 camera mounted on an Olympus SZX12 stereo microscope. Fu used a DP72 CCD camera mounted on a Nikon SZX 16 stereo microscope.

Figure or Figures refer to diagrams in this paper; for abbreviations in text or figures, see Table A1.

### 2.2. Taxonomic Characters

Characters of aedeagus and sheath are presented in each species description on a comparative basis.

The points of articulation of the anterior arms of the aedeagal sheath tergite with the sides of the sheath sternite are compared from beneath, along a horizontal line, as either level with each other or with the right-side arm more anterior than the left.

The median line in larvae runs from the anterior margin of thoracic segment 2 to the posterior margin of abdominal segment 8 and may be narrow, with large closely approximate tergal plates in each segment, except for abdominal segment 9 [13] (Figures 73, 75, and 77), or wide with smaller well-separated tergal plates in each segment, except for abdominal segment 9 [4] (Figure 8). 

### 2.3. Larval Rearing

Five pairs of male and female adults were placed in a Petri dish with moist cotton at room temperature 25 *±* 1 °C with a natural light period. Adults were allowed to mate, and females laid eggs on the moist cotton. Hatched larvae were removed and placed in a Petri dish with water to examine whether larvae had gills and could live in water.

### 2.4. Sample Collection for Molecular Analysis

Specimens collected from different localities were deposited in NHMHAU. 

For genome sequencing, Fu collected additional dozens of individuals of each species that were immediately preserved in anhydrous ethanol, followed by preservation at −40 °C in the laboratory prior to DNA extraction. A single individual was used to extract DNA.

### 2.5. DNA Extraction Sequencing, Assembly, and Annotation of Mitogenomes

Total DNA was extracted from the whole insect body using a modified CTAB method [20]. RNase A was used to remove RNA contaminants. The integrity and concentration of the DNA were verified by 1% agarose gel electrophoresis and Qubit fluorometry. The electrophoresis result showed a clean single-band product, and the DNA concentration was higher than 100 ng/μL. Then, 350 bp short-insert libraries were prepared following Illumina’s instructions. An Illumina Novaseq 6000 (Illumina Inc., San Diego, CA, USA) platform was then employed for whole-genome sequencing according to the standard Illumina protocols. To retain reads of high quality, the following method was used: First, the adaptors were removed from the sequencing reads. Second, read pairs were excluded if any one end had an average quality lower than 20. Third, ends of reads were trimmed if the average quality was lower than 20 in a sliding window size of 5 bp. Finally, read pairs with any end shorter than 75 bp were removed. Clean reads were used to produce a de novo assembly using IDBA-UD [21], with minimum and maximum k values of 41 and 141 bp, respectively. The mitogenome sequences of 3 species of fireflies, namely, *Aquatica qingshen* sp. nov., *Aquatica xianning* sp. nov., and *Aquatica lateralis,* were identified b Geneious 10.1.3 (http://www.geneious.com, accessed on 4 January 2022). Genomic annotations were performed using MITOZ [22] and tRNAscan-SE 2.0 [23].

### 2.6. Phylogenetic Analysis

Mitogenome sequences of the following were downloaded from the NCBI’s nucleotide database (GenBank): *Abscondita anceyi* (Olivier) (MH020192), *Abscondita cerata* (Olivier) (MW751423), *Abscondita terminalis* (Olivier) (MK292092), *Aquatica ficta* (Olivier) (KX758085), *Aquatica lateralis* Japan (LC306678), *Aquatica leii* (Fu *et* Ballantyne) (KF667531), *Aquatica wuhana* Fu *et* Ballantyne (KX758086), *Asymmetricata circumdata* (Motschulsky) (KX229747), *Curtos bilineatus* Pic (MK292114), *Curtos costipennis* (Gorham) (MK609965), *Curtos fulvocapitalis* Jeng Yang et al. (MW582616), *Nipponoluciola cruciata* (Motschulsky) (AB849456), *Luciola curtithorax* Pic (MG770613), *Pteroptyx maipo* Ballantyne (MF686051), *Pygoluciola qingyu* Fu et Ballantyne (MN688374), *Pyrocoelia rufa* Olivier (AF452048), *Sclerotia flavida* (Hope) (KP313820), *Tribolium castaneum* (NC_002081). PhyloSuite v1.2.2 [24] was used to extract 13 PCGs, 22 tRNA genes, and 2 rRNA genes. MAFFT [25] was used to make multiple sequence alignments for extracting genes and to optimise the 13 PCGs’ codon alignments using MACSE v. 2.03 [26], which preserves the reading frame and allows for the incorporation of sequencing errors or sequences with frameshifts. In order to remove misaligned sites, as well as substitutionally saturated sites, from the alignments, Gblocks [27] was used for PCGs with the following parameter settings: minimum number of sequences for a conserved/flank position (11/11), maximum number of contiguous non-conserved positions (5), minimum length of a block (4), and allowed gap positions (with half). trimAl [28] was used for amino acid sequences of PCGs and RNA genes using the “-automated1” command.

The pairwise genetic distances (p-distance) between all taxa in this study were calculated using ABGD web (https://bioinfo.mnhn.fr/abi/public/abgd/abgdweb.html, accessed on 7 December 2023) for species delimitation with the Kimura two-parameter (K80) distance model.

All aligned genes were concatenated into three datasets using PhyloSuite v1.2.2.: (1) the 13 amino acid sequences of all PCGs (AA) were combined; (2) the 13 nucleotide sequences of all protein-coding genes (PCGs) were combined; and (3) the PCGs, 22 tRNAs, and 2 rRNA genes (PCGsRNA) were combined. ModelFinder [29] was used to select the best-fit model following the BIC criterion in the three generated datasets for IQ-TREE and MrBayes, respectively. The best-fit models for AA were mtMet+F+R4 and mtREV+F+I+G4, and those for PCGs and PCGsRNA were GTR+F+R4 and GTR+F+I+G4. 

Topologies on the datasets were compared using the phylogenetic methods of maximum likelihood (ML) and Bayesian inference (BI). *Tribolium castaneum* was used as outgroup. IQ-TREE [30] was used to perform ML analysis, and bootstrap support (BS) was assessed using 5000 ultrafast [31] bootstrap replicates. MrBayes 3.2.6 [32] was used to perform a BI analysis. The analyses of each dataset were performed with 4 MCMC chains and run for 20 million generations. Every 1000th generation was sampled as a consensus tree. The type of consensus tree was Halfcompat. The convergence of the independent runs was indicated by a standard deviation of split frequencies <0.01 and an estimated sample size (ESS) > 200. The initial 25% of sampled data were discarded as burn-in data, and the remaining trees were used to represent the values of posterior probability (PP).

The p-distance was calculated using ASAP for species delimitation with the Kimura two-parameter (K80) distance model [33].

### 2.7. Data Availability

The three newly sequenced mitogenomes were submitted to the GenBank database under the accession numbers of *Aquatica qingshen* sp. nov. (OM135505), *Aquatica xianning* sp. nov. (OM135504), and *Aquatic lateralis* (Chinese population, OM135506).

## 3. Results

### 3.1. Outcomes

The number of species now known in *Aquatica* has increased to seven, with all but one known from the features of males, females, and larvae. The aquatic association has been confirmed for all, although we were unable to breed some larvae past the first instar. A conventional morphological taxonomy is combined with molecular information, an uncommon occurrence thus far within the Luciolinae (see [34]). Both morphological and molecular phylogeny confirm their distinct species status (except for *A*. *leii* and *A. ficta,* which are discussed further), as well as the definition of the genus *Aquatica*. 

Almost all species are now characterised by the features of the female reproductive system, which possesses a distinctively shaped MOP. While in larvae, the wide median line between most tergal plates is seen in both species of *Nipponoluciola*, it is also a feature of *Aquatica hydrophila*, and this feature may be due in part to dehydration or the method of preservation. The narrow line characterises the remaining *Aquatica* species. Other features for which we currently do not have enough information, like the number of pygopods, the number of their basal branches, and the nature of the secretory glands [35], could provide additional information.

Insects known to have an association with bodies of water are often threatened species if those water bodies dry up. Assessing such threats requires knowledge of the life cycles, especially the ability to name the species. It is however the classification of just what constitutes an aquatic insect that varies within the Lampyridae, especially since the aquatic lifestyle may not relate to both adults and larvae. The definitions of aquatic status or otherwise are modified to specifically address the Luciolinae, where only the larvae are aquatic.

### 3.2. Categories of Aquatic Luciolinae Larvae

Combining the categories of Fu et al. [13] and those of Jӓch [15] for Luciolinae larvae with an association with water, the following definitions are used:Aquatic: Conforms with Jӓch’s [15] category of false water beetles, where the adults are terrestrial and free-flying, and larvae live under water and have obvious adaptations for aquatic life. Aquatic larvae exist in two forms:
Abdominal gills: those that live in shallow usually well-oxygenated water; with tracheal gills and spiracles along the margins of most abdominal segments; very soft bodied; with 10 pairs of defensive organs along the sides of the meso- and metathorax and abdominal segments 1–8, which, when everted, emit odours that can be characteristic of the species. This form includes *Aquatica* spp. and *Nipponoluciola* spp. [4].Metapneustic: those that swim upside down, just beneath the water surface; no gills; respiration is by paired spiracles on the dorsal surface of abdominal segment 8. Hard-bodied and no obvious defensive organs. This form includes *Sclerotia* sp. [13].
Semiaquatic: Conforms to Jӓch’s [15] (p. 28) category of facultative aquatic beetles; Jӓch indicated that they “occasionally (or regularly) and actively stay submerged for a limited period (for hunting, feeding, seeking refuge) in any of their developmental stages. There are no conspicuous morphological adaptations for aquatic life.” In contrast, the semiaquatic Luciolinae may live close to water, but the adults do not submerge. The following have semiaquatic larvae: *Pygoluciola qingyu* [3], *Pteroptyx* sp. [34], and *Atyphella aphrogeneia* [14].

### 3.3. List of Luciolinae genera with Aquatic Larvae

Three *Luciolinae genera* with aquatic larvae are known: *Aquatica* with five species, *Sclerotia* with three, and *Nipponoluciola* with two [4]. Two new species of *Aquatica* are listed in Appendix A.

### 3.4. Key to Genera with Aquatic Larvae Using Males

A key to the genera *Aquatica, Nipponoluciola,* and *Sclerotia*, using male characters is in [4].

### 3.5. Key to Aquatic Larvae with Gills in the Genera Aquatica and Nipponoluciola

Median line wide [16] (Figure 39), [4] (Figures 8 and 9); coloured tergal plates widely separated in median line……………………………………………………………..........**2**;


Median line narrow [1] (Figures 25 and 30); coloured tergal plates closely approximate in median line……………………………………………...……………………...….**4**;

2.Protergum without marginal pale markings [36] (p. 165 *Luciola* sp. 1 is *A. hydrophila*), [16] (Figure 3); central coloured plate with arcuate lateral margins; paired coloured dorsal plates on body segments 2–11 without paler markings……………………………………………………………..***A. hydrophila* (Jeng et al.)**;


All terga, except for terminal abdominal tergum, with some pale lateral margins; central coloured protergal plate of varying shape; paired coloured dorsal plates on body segments 2–11 with paler markings along lateral margins; terminal tergum with pale posterior margin ……………………………………………………….............**3**;

3.Protergum with entire ovoid dark median marking narrowly encircled by a pale margin [4] (Figures 8A and 9)……………..….......***Nipponoluciola cruciata* (Motschulsky)**;


Protergum with entire dark median marking assuming a cross-like shape, with four discrete pale areas at anterolateral and posterolateral corners [4] (Figures 8B and 9)……………………………………….……………………….***N. owadai* Satô et Kimura**;

4.Protergum with two pale markings only at anterolateral corners [1] (Figures 25 and 26)……………………………………………………………....***A. leii* Fu et al. Ballantyne**;


Protergum with more than two pale markings around margins....................................**5**;

5.Protergum subparallel-sided; with four separated pale marks at anterolateral corners and along lateral margin anterior to the posterolateral corners; posterior margin dark-coloured………………………………………………...***A. lateralis* (Motschulsky)**;


Protergum with lateral margins not subparallel-sided, arcuate; having four discrete pale markings at anterolateral and posterolateral corners; always with some pale-coloured markings along posterior margin……………………………………………...**6**;

6.Protergum mainly dark brown to black, with four discrete pale-coloured areas at anterolateral and posterolateral corners and a wide dark brown to black area along middle of posterior margin; terminal tergum with lateral margins diverging posteriorly [13] (Figures 73 and 74)…………………………………………………***A. ficta* (Olivier)**;


Protergum mainly dark brown to black with extensive pale-coloured markings obliquely across both posterolateral corners and across posterior margin, except for very narrow median black marking; terminal tergum with lateral margins wider at anterior and posterior ends [1] (Figures 30 and 31)…...***A. wuhana* Fu et al. Ballantyne**.

### 3.6. Taxonomy: The genus Aquatica with Two New Species from China

*Aquatica* Fu et al. 2010

Figure 1, Figure 2, Figure 3, Figure 4, Figure 5, Figure 6 and Figure 7, Appendix A

**Diagnosis**. The generic diagnosis in [3] is modified and abridged here. The two new species are known from China, with *A. lateralis* also known from Japan. Free-flying adults usually have dark brown to black elytra, often narrowly pale-margined, and orange-yellow pronotum, often with median dark markings, and they are found near bodies of water. Associated larvae are aquatic and soft-bodied, with eight pairs of forked lateral abdominal gills and spiracles along the sides of the abdomen. One of only three genera of *Luciolinae* having aquatic larvae (with the others being *Nipponoluciola* Ballantyne et al. and *Sclerotia* Ballantyne), *Aquatica* is one of a large group of genera where the aedeagus has LL widely visible beside the ML; within that group, it belongs to a group of 11 genera (*Atyphella* Olliff, *Aquilonia* Ballantyne, *Convexa* Ballantyne, *Emeia* Fu et al., *Lloydiella* Ballantyne, *Missimia* Ballantyne, *Pacifica* Ballantyne, *Pygatyphella* (Ballantyne), *Sclerotia* Ballantyne, and *Triangulara* Pimpasalee), where the posterior portion of the male aedeagal sheath sternite is emarginated on the right. *Aquatica* is distinctive in this group, as the posterior half of sternite has a median, slightly oblique, ridge, and is irregularly emarginated along both sides, with each margin having at least one pointed tooth. It is most obviously distinguished from *Sclerotia* by the absence of the following features: sclerites in a band of muscle surrounding the aedeagal sheath, parallel-sided elytral punctation, and metapneustic aquatic larvae [2] (Figures 29–31). Adult males of *Aquatica* are distinguished from *Nipponoluciola* by the nature of the elytral epipleuron in the adult [4] (p. 15, Figure 3) (Appendix A).

Females are macropterous, have been observed in flight, and are coloured like males, and those investigated have paired elongated slender bursa plates and a well-defined MOP at the opening of the median oviduct (Appendix A). 

Larvae have eight pairs of branched abdominal gills and are currently indistinguishable from the larvae of *Nipponoluciola,* except by dorsal colour variations.

**Generic description**. We expand the brief generic treatment of the female given in [1].

**Female**. Macropterous. Colour: coloured like male (males usually have dark brown to black elytra, orange-yellow pronotum; several species have dark brown, narrowly pale-margined elytra and median dark pronotal mark; one species has pale yellowish-brown elytra). Body outline: similar to that of male; pronotal width slightly greater than humeral width; elytra parallel-sided and contiguous along their sutural margins for almost all of their length; interstitial lines 1 and 2 visible but not well elevated. Head: well-developed mouthparts, assumed capable of feeding as an adult; apical labial palpomeres flattened, subtriangular in outline, with inner (longer) margin dentate; antennae with all segments elongated, slender. Legs: no legs with curved or swollen femora or tibiae. Abdomen: posterior margin of V7 emarginated; terminal two segments not heavily sclerotised or projecting beyond elytral apices unless the female is gravid; anterior margin of V8 continuous with median apodeme, well sclerotised, separated from darker posterior portion of V8 by clear membranous section (Figure 3n, Figure 5m,n, Figure 6b, and Figure 7l,m). Reproductive system: elongate slender bursa plates with pointed posterior margin; MOP well defined with variable shape, either restricted to the median oviduct, or elongated and moderately wide, with a wide anterior margin (extending into the median oviduct in two species, Figure 1g,h, Figure 3k–m, Figure 4, Figure 5m–o, Figure 6, and Figure 7l,m) [1] (Figure 1b plate is arrowed). SDG observed in several species, and spermatophore identified within the bursa in several species (Appendix A). Slender FAG observed in three species.

**Remarks**. Species now assigned to *Aquatica* were illustrated as species of *Luciola* in Chen [36] (p. 163 depicts *L. ficta*; p. 164 depicts a copulating pair of *L. hydrophila*). The egg-laying behaviour of *L*. *ficta* was described in [16]. The female of *Luciola ficta* was described in [37], including details of the reproductive system and illustrations of an intact spermatophore in the SDG, thin needle-like plates in the sides of the bursa, and a prominent elongated rectangular outline of a MOP. It is not possible from this diagram to see whether this plate also extends into the wall of the vagina (this is addressed below). Two possible functions for the MOP were suggested: (1) it prevents the spermatophore from being inserted into the median oviduct or (2) pressure from the plate on the spermatheca as an egg traverses the median oviduct could cause the release of sperm [1]. 

Some interpretations [37] differ, describing *L. ficta* (sic) with a “sclerotized plate located in the bursal wall posterior to the common oviduct” (we define the bursa as anterior to the median oviduct) and *L. ficta* lacking the “needle like sclerotized plates that were embedded in the bursal walls of *L. lateralis* females” [37] (Figure 3c,d) (these plates are present in *ficta*). The “sclerotized plate located in the bursal wall” in *L. ficta* is probably the MOP, which was observed by us in both the vaginal wall (bursal wall in [37]), posterior to the entry of the common oviduct, and in the wall of the common oviduct, where it joins the vagina. *A. ficta* has paired needle-like bursa plates (Figure 1g,h). Figure 1b in [37] indicates a “sclerotized plate” narrow at its anterior end and expanded and slightly bifurcate at its posterior end, and it is the MOP of *A. lateralis* (Figure 3l,m). The projection from the spermatophore into the base of the spermathecal duct was first observed in [37] (Figure 2d) and called the thumb. The authors identified a “sclerotized plate located in the bursal wall posterior to the common oviduct”, which is the MOP (here, we refer to this area as the vagina), but they did not see bursa plates in *A. ficta*.

#### 3.6.1. Key to Species of *Aquatica* Fu et al. 2010 Using Males

This key expands on Fu et al. [1] (p. 5):
Pronotum usually with some darker-coloured median markings; MS often dusky brown or black; ML often strongly curved, apically rounded, and expanded; apex of aedeagal sheath sternite entire or split; LL not hooked at their tips, or if LL hooked at their tips, then ML strongly curved and apically expanded; teeth often present on internal dorsal margins of LL; mid-anterior margin of basal piece thickened, often expanded to either side of median line; basal piece often in two laterally expanded pieces; posterior half of sheath sternite strongly emarginated on each side ………….**2**;

Pronotum orange or yellow with no darker-coloured markings; MS usually pale light brown to yellow; ML not strongly curved or apically rounded and expanded; apex of aedeagal sheath sternite entire; LL elongated slender, hooked, or inturned at their tips; no teeth on internal dorsal margins of LL; mid-anterior margin of basal piece not thickened or expanded; aedeagal sheath sternite not strongly emarginated on each side of posterior half ………………………………………………………………..**5**;
2.Aedeagal ML strongly curved and apically expanded in vertical plane; LL strongly divergent along inner dorsal margins; basal piece with wide halves; tip of sheath sternite split into two truncated pieces (Figure 5g–k and Figure 7g–k) ……………..………**3**;

Aedeagal ML not strongly curved or apically expanded; LL not strongly divergent along inner dorsal margins; basal piece with narrow halves; tip of sheath sternite entire and rounded, not split into two truncated pieces…………………………………..**4**;
3.Large; 8–12 mm long; basal piece not subdivided; mid-anterior margin of basal piece thickened and expanded ………………………………….. ***xianning* sp. nov. (Figure 7)**;

Smaller; 5.3–6.3 mm long; basal piece subdivided; mid-anterior margin of basal piece not thickened or expanded …………………………………***qingshen* sp. nov. (Figure 5)**;
4.Pronotum with wide median dark brown marking extending from anterior to posterior margins (Figure 3a); MS black; LL with inner basal area toothed; basal piece not narrowly sclerotised along anterior margin and does not bear two fine rounded projections (Figure 3) [1] (Figures 27–37); V7 with black markings posterior to LO [1] (Figures 8–15); aedeagal sheath with lateral junctions of tergite arms and sternite margins not level (viewed from beneath the right junction is anterior to the left) (Figure 3h,i); posterior area of sheath sternite narrow, strongly emarginated on both sides (Figure 3h,i)……………………………………………………...***lateralis* (Motschulsky)**;

Pronotum with anterior median dark brown marking not extending to posterior margin; MS pale yellow or dusky brown, never black; inner basal area of LL not toothed; basal piece narrowly sclerotised along anterior margin which bears two fine rounded projections [1] (Figures 27–37); V7 without black markings posterior to the LO; aedeagal sheath with lateral junctions of tergite arms and sternite margins level (viewed from beneath the right junction is level with the left); posterior area of sheath sternite wide, not strongly emarginated on both sides …………...***wuhana* Fu et al. Ballantyne**;
5.Known from Taiwan only; elytra very dark brown, almost black, without paler margins; aedeagus short and wide (L/W 2.0); LL widely expanded in horizontal plane, 3–4 X as wide as ML apex; apices not narrowed or inturned …………………………………………………***hydrophila* (Jeng et al.) (Figure 2)**;

Recorded from both Taiwan and mainland China; elytra either dark brown with paler margins or yellowish; aedeagus elongated and slender (L/W 3.7); LL not widely expanded in horizontal plane; apices narrowed and slightly inturned…………………**6**;
6.Dorsal surface yellow-pale brown, margins concolourous; V5 dark brown; V3, 4 yellow [1] (Figures 17–26)…………………………………………….***leii* (Fu et Ballantyne)**

Dorsal surface elytra dark brown with all margins, except for base, narrowly yellow; all abdominal ventrites anterior to the LO very dark-coloured, almost black [1] (Figures 2–7)…….………………………………………………………………...***ficta* (Olivier)**.


#### 3.6.2. Key to Species of *Aquatica* Fu et al. 2010 Using Females

Pronotum pale yellow with no darker markings………………………………………..**2**;


Dorsal surface always with dark brown elytra, which may be pale-margined; pronotum usually with median darker markings………………………...……………………**4**;

2.MOP elongated slender subparallel-sided extending into both median oviduct and vagina (Figure 4)……………………………………………………………………………**3**;


MOP structure unknown; elytra black; pronotum yellow without darker markings (Figure 2c,d)…………………………………….…………………***hydrophila* (Jeng et al.)**;

3.Dorsal surface yellowish without darker markings on elytra; V4, 5 black………………………………………………………...…...***leii* (Fu et al. Ballantyne)**;


Elytra dark brown with narrowly pale suture, apex, and lateral margin; V2–5 black…...…………………………………………………………………...…***ficta* (Olivier)**;

4.MOP elongated much longer than wide (Figures 3k–m and 5c,d)……………...……...**5**;


MOP about as long as wide (Figures 5m–o and 7l,m)………………………………….**6**;

5.Pronotum reddish with median dark marking extending from anterior to posterior margins; MOP elongated slender with expanded and emarginated posterior apex; (Figure 3c,d,k–n) ………………………………………...……...***lateralis* (Motschulsky)**;


Pronotum yellowish orange with anteromedian dark marking not extending to posterior margin; MOP elongated, longer than wide, posterior area with margins converging to a broadly rounded apex, median anterior margin narrowly produced, sides expanding (Figure 6c,d)…………………….…………...***wuhana* Fu et Ballantyne**;

6.7–7.1 mm long; MOP about as wide as long, with anterior margin entire and rounded (Figure 5e,k–m)………………………………………………………….***qingshen* sp. nov.**;

10–13 mm long; MOP about as wide as long, with short truncated median projection, slightly curved sides, and emarginated posterior margin (Figure 7e,f,l,m) ………………………………………………………….………..***xianning* sp. nov**.


***Aquatica ficta* (Olivier 1909)**


**Figure 1 insects-15-00031-f001:**
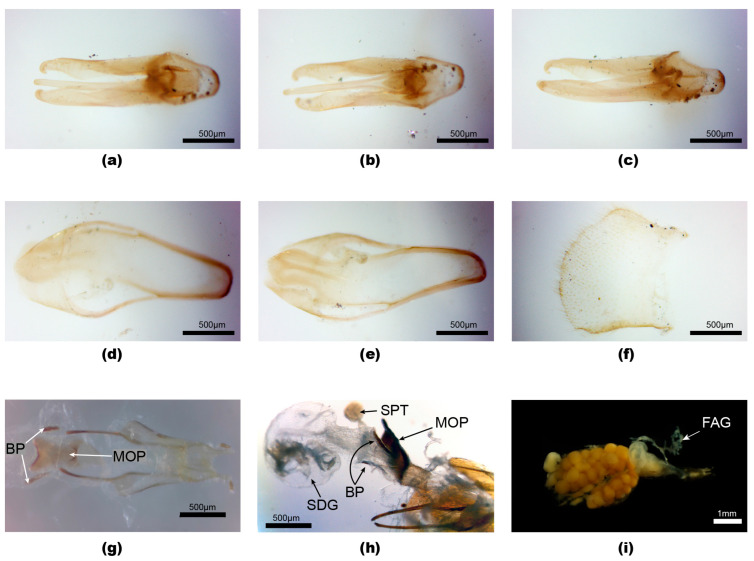
*Aquatica ficta**.*** Male (**a**–**f**), female (**g**–**i**). (**a**–**c**) Aedeagus: (**a**) dorsal; (**b**) ventral; (**c**) slightly oblique dorso-lateral. (**d**,**e**) Aedeagal sheath: (**d**) dorsal; (**e**) ventral. (**f**) Tergite 8. (**g**–**i**) Female reproductive system: (**g**) ovipositor with median oviduct plate and bursa plates; (**h**) cleared system with SDG to left; (**i**) intact reproductive system with female accessory gland. Panels (**a**–**f**) with anterior end to right of page, (**g**–**i**) with anterior end to left of page. Legend: BP, bursa plates; FAG, female accessory gland; MOP, median oviduct plate; SDG, spermatophore-digesting gland; SPT, spermatheca.

**Lectotype.** Female**. CHINA.** (MNHN). Not examined by these authors.

**Specimens examined**. **CHINA**. Collector is Fu, X. H. Guizhou Province, Bijie City, 1 male 2 females, 23. vii. 2016; Guangdong Province, Meizhou City, 3 males 2 females, 13. vi. 2018; Guangdong Province, Huizhou City, 2 males, female, 8. vi. 2020; Jiangxi Province, Ganzhou City, Shangyou County, 3 males 1 female, 20. vi. 2021; Sichuan Province, Meishan City, Qingshen County, 4 males, 1 female, 27. V. 2022; Fujian Province, Fuqing City, 4 females, 14. iv. 2008; Hubei Province, Xianning City, Tongshan County, Xiapu Town, 5 males, 3 females, 18. v. 2018. (NHMHAU). Hubei Province, Wuhan City, 3 females bred in laboratory (ANIC).

**Diagnosis**. Male: *A. ficta* belongs to a group of two species with a pale yellow pronotum, an aedeagal sheath sternite with an entire apex, and an elongated slender aedeagus (L/W 3.7), with LL apices inturned, appearing hook-like (Figure 1a–c). It is distinguished from *A. leii* by the black V2–5 anterior to the LO, and the dark brown, pale-margined elytra (*A. leii* has V3, 4 yellow; V5 brown; and yellowish-brown elytra) [1] (Figures 2–4, 17, and 18). Female: coloured like the male with an orange pronotum, dark brown elytra with narrow pale orange margins extending along suture, lateral margins, and very narrowly around apex; with V7, 8 being pale yellowish, semitransparent; with an elongated parallel-sided MOP extending into both the vagina and median oviduct (similar to Figure 4). 

**Male genitalia** [1] (Figures 5–7). *Aedeagal sheath*: (Figure 1d,e) sheath sternite L/W 3.7, with base wide and broadly rounded, apex entire, rounded, a single pointed projection along each margin in posterior area, and broad posterior half of sternite; R tergite arm joining the side of sternite anterior to that of L tergite arm. *Aedeagus:* (Figure 1a–c) L/W 3.7; basal piece incompletely subdivided, anterior margin narrowly dark, not well sclerotised, median anterior area not expanded, not truncated; LL subparallel-sided along most of their length, divergent along the inner dorsal margins, with narrow inturned apices appearing hook-like; ML slender, slightly wider in basal 1/5, not strongly curved when viewed from side, apex rounded, expanding slightly in dorso-ventral plane only.

**Female.** Coloured like male with orange pronotum, dark brown elytra with narrow pale orange margins extending along suture, lateral margins, and very narrowly around apex; ventral abdomen very dark brown, except for white LO in V6, and pale yellowish to yellowish-white, semitransparent, V7, 8 (anterior area of V7 may have underlying fat body) [1] (Figure 4); T7, 8 pale yellow, semitransparent in Fuqing specimen, pale brown in Taiwan specimen; median posterior margin of V7 very shallowly emarginated, that of V8 rounded; anterior apodeme of V8 well sclerotised and separated from darker posterior portion of V8 by clear membranous section. *Reproductive system*: (Figure 1g–i) bursa with slender elongated paired plates; MOP with wide anterior end, projecting into vagina at posterior end. A FAG opens into the bursa.

**Larvae**. Illustrations of larvae included the terminal sense organs on the apical maxillary palpomere and the eversible organs associated with the abdominal tracheal gills [13] (Figures 1–12, 67, 73, and 74).

**Remarks.** Females of *Aquatica ficta* are monandrous [37]. Given that the transfer of a spermatophore to the female provides a longevity benefit for the female, it would be expected that selection would favour polyandry. The monandrous condition was unexpected. The time of transfer of the spermatophore to the female and its subsequent digestion were investigated [37]. As disintegration was complete by 24 h post-mating, persistence of an intact spermatophore in the female reproductive tract could not be considered as an explanation for monandry. Artificial light of certain wavelengths influenced the flash signals of *A. ficta* [38].

*Luciola ficta* was described [39] on the basis of three males from “Kouy-Tcheou, région de Pin Fa” in China, the abdomen having “tribus ultimis ventris segmentis albidis” (last three segments of the abdomen being white), with the remaining segments being black—“l’exception des trois derniers segments, noirs” (the ventral abdomen black, except for the last three segments). This description is inconsistent with the male abdomen depicted above (terminating with two white light organ segments). A lectotype designation of a female [16] made to conserve the taxonomy may not conform to ICZN rules [40]. These inconsistencies are addressed in the Discussion. 


***Aquatica hydrophila* (Jeng et al. 2003)**


**Figure 2 insects-15-00031-f002:**
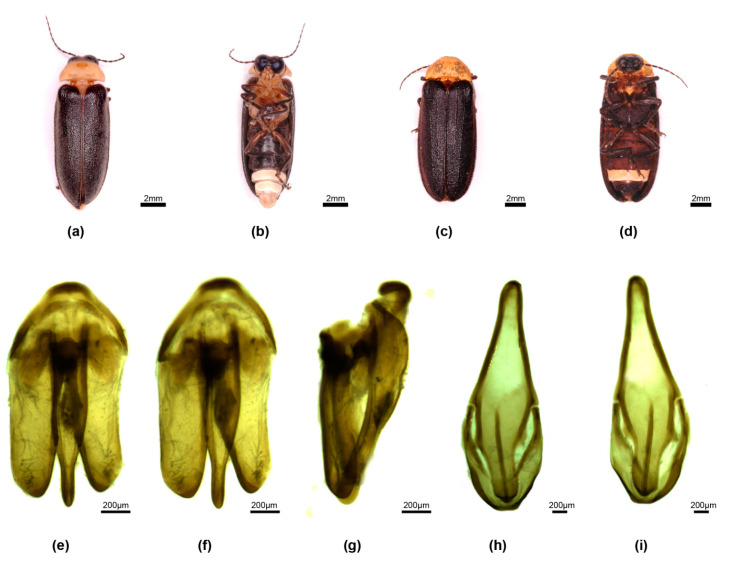
*Aquatica hydrophila.* Male (**a**,**b**,**e**–**i**), female (**c**,**d**). (**a**,**c**) Dorsal habitus; (**b**,**d**) Ventral habitus. (**e**–**g**) Aedeagus: (**e**) ventral; (**f**) dorsal; (**g**) right lateral. (**h**,**i**) Aedeagal sheath: (**h**) dorsal; (**i**) ventral.

**Type.** Male. **CHINA** (Taiwan). NMNS Taiwan.

**Specimens examined. CHINA** (Taiwan). Male, female Ho, J.Z. (NMNS).

**Diagnosis**. *A. hydrophila* is distinctive among *Aquatica* species; males have black elytra without pale margins, a pale orange-yellow and unmarked pronotum, an entire rounded apex of the aedeagal sheath, and a short squat aedeagus (L/W 2.0) (Figure 2). Females are coloured like males, with venter being almost black, except for white LO in V6. 

**Male** (features additional to Jeng et al. 2003).

**Male genitalia**. *Aedeagal sheath*: (Figure 2h,i) sheath sternite L/W 3.2, base narrowly rounded, posterior half narrows irregularly to a slightly acute, entire apex; a single pointed projection along each side of emarginated posterior section, posterior section narrow (width across narrowest part of apex/width across tergite at same point 0.33); both R and L tergite arms join sides of sternite equidistant from sternite base. *Aedeagus*: (Figure 2e–g) L/W 2/1; basal piece not clearly subdivided, anterior margin thickened, darker than rest, mid-anterior margin appearing darker and slightly thicker than rest but not otherwise differentiated; LL subparallel-sided (outer margin of R lobe straight along most of its length, that of L slightly curved), apices broadly rounded, not inturned; ML not strongly curved, not projecting strongly beyond LL when viewed from side, subparallel-sided, wider in basal 2/3, narrowing in apical 1/3, apex not expanded in dorso-ventral plane.

**Female**. Reproductive system not investigated.

**Larva**. A coloured plate of the larva depicts its dorsal surface, and the aquatic nature was confirmed [16].

**Remarks**. This species was briefly redescribed and figured [1]. We present here additional features illustrating aspects of male morphology, but we were unable to obtain females for dissection.


***Aquatica lateralis* (Motschulsky 1860)**


**Figure 3 insects-15-00031-f003:**
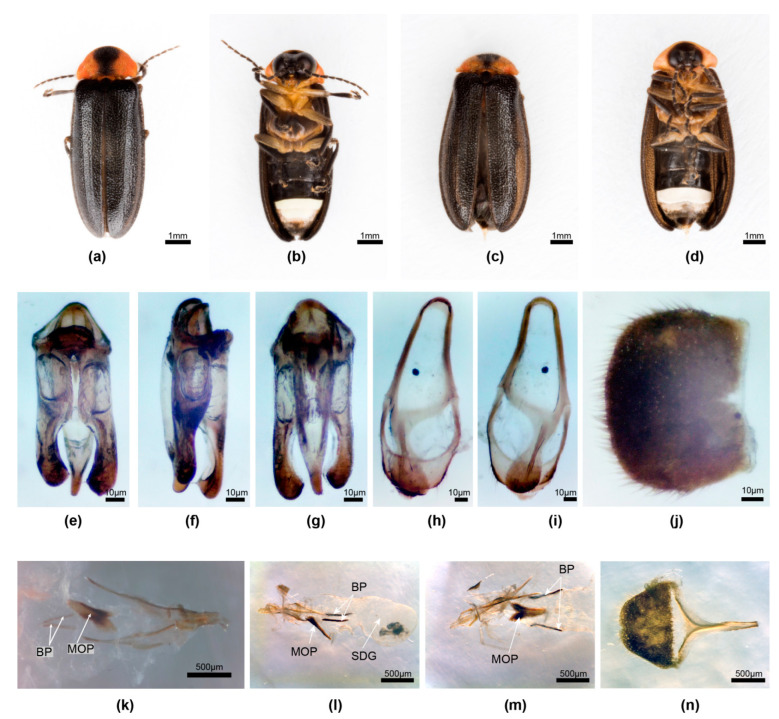
*Aquatica lateralis.* Male (**a**,**b**,**e**–**j**), female (**c**,**d**,**k**–**n**). (**a**,**c**) Dorsal habitus; (**b**,**d**) Ventral habitus. (**e**–**g**) Aedeagus: (**e**) dorsal; (**f**) slightly oblique left lateral; (**g**) ventral. (**h**,**i**) Aedeagal sheath: (**h**) dorsal; (**i**) ventral. (**j**) Tergite 8. (**k**–**m**) Female reproductive system: (**k**) ovipositor with bursa, BP and MOP; (**l**) complete reproductive system with SDG to right; (**m**) detail of MOP and BP from below. (**n**) Tergite 8. Panel (**k**) anterior end to left of page; (**l**–**n**) anterior end to right of page. Legend: BP, bursa plate; MOP, median oviduct plate; SDG, spermatophore-digesting gland.

**Type. RUSSIA.** Dahourie. ZMMU. Not examined by these authors.

**Specimens examined**. **CHINA.** Collector is Fu, X. H. Liaoning Province, Dandong City, 3.vi. 2021, 8 males, 3 females (NHMHAU). **JAPAN**. Female originating from firefly populations in Fukushima and Tochigi, Honshu, and reared by Dr. Norio Abe at the Firefly Breeding Institute in Itabashi Ward, Tokyo, kindly supplied by Adam South (ANIC). Collector is Ohba, N. Yokosuka City, Kanagawa Prefecture., 1 male, 20. vi. 1982. (ANIC). Collector is Ohba, N. Matsuyama City, Ehinc Prefecture., 1 male, 8. vii. 1996. (ANIC). Tokyo Metropolitan University, Minami Ohsawa, Hachioji, Tokyo, 3 males, 28. v. 2008 (NHMHAU). 

**Diagnosis**. Male: *A. lateralis* males are distinctive because of their pinkish pronotum, having a wide median dark brown marking extending from anterior to posterior margins, black MS, black markings on V7 posterior to LO, and aedeagal LL closely approximate at their bases dorsally, with inner margins toothed. Female: Coloured like male with very dark, uniformly brown elytra, and pinkish orange pronotum having median dark marking extending from anterior to posterior margin, V7 semitransparent with anterior half whitish due to underlying fat body, semitransparent posterior half of V7 appearing black due to underlying black V8; MOP well defined, tapering to an acute anterior end with posterior end bifurcate, not obviously extending into vagina. 

**Male genitalia** [1] (Figures 10–14). *Aedeagal sheath*: (Figure 3h,i) sheath sternite L/W 2.6/1, base wide, broadly rounded, apex very shallowly emarginated, a single pointed projection along each margin in posterior area, posterior half of sternite much narrower than anterior portion; R, L tergite arms join the side of sternite approximately equidistant from base of sternite. *Aedeagus:* (Figure 3e–g) L/W 3.0/1; basal piece incompletely subdivided; anterior margin narrowly dark, not well sclerotised, median anterior area slightly expanded, truncated; LL subparallel-sided along most of their length, L lobe slightly longer than R, inner dorsal margins toothed, closely approaching in basal 1/3, with wide inturned apices appearing hook-like; ML expanded along most of length, narrowing in apical 1/10, not strongly curved when viewed from the side, with apex rounded and expanding slightly in dorso-ventral plane only.

**Female**. (Figure 3c,d,k–n) Segments 7 and 8 of the gravid abdomen may protrude beyond elytral apices. Colour: like male, with pinkish-orange pronotum having median dark brown marking extending from anterior to posterior margins; MS, mesonotal plates almost black; elytra very dark brown without paler margins; ventral body almost black, except for pale white LO in V6, pinkish fat body occupying almost all of V7, except for a narrow semitransparent posterior band, and dark brown V8; T7 mid-brown, T8 very dark brown; V2 clearly divided into two; anterior apodeme and anterior margin of V8 pale, semitransparent, well sclerotised, separated from the darker posterior area by a pale-coloured membranous portion. *Reproductive system*: (Figure 3k–m) bursa with well-defined elongated slender paired plates; MOP well defined, tapering to an acute anterior end with posterior end bifurcate, not obviously extending into vagina.

**Remarks.** This is the first discovery of an apparent *A. lateralis* population in China, which mitogenome sequencing indicated differs from the Japanese population. Future work will first focus on larval collection to confirm aquatic status. Larvae have not yet been discovered for this population but are reliably associated for the Japanese populations [4].

*Luciola lateralis* was originally recorded in Russia and Japan [41]. Fu collected the specimens addressed here from Dandong city, a border city of China close to North Korea, in 2021. We follow the interpretation of *Luciola lateralis* from Japan [16] in the absence of type material. Larvae are found in rice paddies [16], and males can copulate with a Taiwanese firefly that may be *ficta* [42,43,44]. The species identity and confusion over taxonomic treatments were addressed [4].

“Needle like sclerotized plates that were embedded in the bursal walls of *L. lateralis* females” were observed [37] (Figures 1b and 3c,d), as well as a “sclerotized plate” narrow at its anterior end and expanded and slightly bifurcate at its posterior end, identified here as the MOP of *A. lateralis*.

The species is widely distributed in Japan, Siberia, and the Korean peninsula [4], and it is revered as the Korean native firefly [45] and the Heike-botaru in Japan [4]. Differing genetic makeup between these populations was recorded [6].


***Aquatica leii* (Fu *et* Ballantyne 2006)**


**Figure 4 insects-15-00031-f004:**
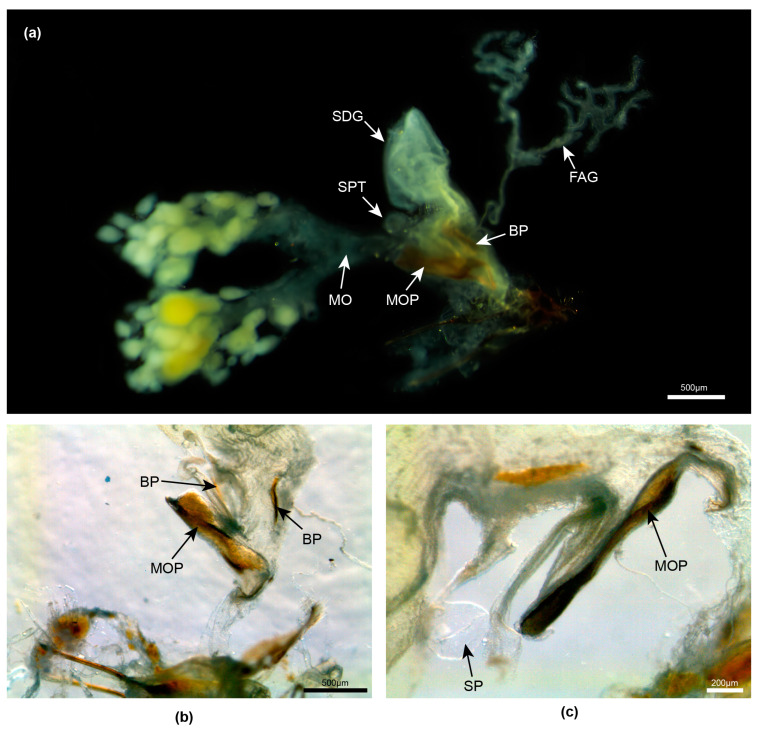
*Aquatica leii.* Female reproductive system. (**a**) Entire system before clearing; (**b**,**c**) detail junction of median oviduct with vagina. Legend: BP, bursa plate; FAG, female accessory gland; MO, median oviduct; MOP, median oviduct plate; SDG, spermatophore-digesting gland; SP, spermatheca.

**Specimens examined**. CHINA. Collector is Fu, X. H. Hubei Province, Huangshi City, Wangying Village, Yangxin County, 1 male, 1 female, 18. v. 2013. (ANIC). Hubei Province, Huangshi City, Yangxin County, Xiandao Lake, 2 males and 1 female, 27. v. 2016; Hubei Province, Ezhou city, Liangzihu District, 5 males, 2 females, 26. v. 2023; Zhejiang Province, Hangzhou city, Fuyang District, 6 males, 2 females, 12. v. 2015; Shandong Province, Ji’nan City, Laiwu District, 2 males, 1 female, 25. v. 2017; Zhejiang Province, Quzhou city, Longyou County, 3 males, 2 females, 28. v. 2023. (NHMHAU).

**Diagnosis**. Male: *A. leii* is one of two species with pale orange yellow pronota, aedeagal sheath sternite apex entire, elongated slender aedeagus (L/W 3.7), and LL appearing hooked at their tips. It is distinguished from *A. ficta* by the black abdominal ventrites anterior to the LO, and the dark brown, pale yellowish margined elytra (*A. leii* has V5 brown; V3, 4 yellow; and yellowish-brown elytra). Female: Coloured like male with yellowish orange pronotum and very light yellowish-brown elytra. 

**Male genitalia** [1] (Figures 19–24). *Aedeagal sheath*: sheath sternite L/W 3.7, base wide, broadly rounded, apex entire, rounded, a single pointed projection along each margin in posterior area, posterior half of sternite broad; R tergite arm joins the side of sternite anterior to that of L tergite arm. *Aedeagus:* L/W 3.7; basal piece incompletely subdivided, anterior margin narrowly dark, not well sclerotised, median anterior area neither expanded nor truncated; LL subparallel-sided along most of their length, divergent along inner dorsal margins, with narrow inturned apices, which may appear hook-like; ML slender, slightly wider in basal 1/5, not strongly curved when viewed from the side, with apex rounded and expanding slightly in dorso-ventral plane only.

**Female**. (Figure 4) Coloured like the male. Here, we provide photos of features of the reproductive system and establish that the MOP is elongated and projects into the vagina at its posterior end. A filamentous branched FAG opens into the bursa.

**Larva**. Aspects of the larvae, including the terminal sense organs on the apical maxillary palpomere and the eversible organs associated with the abdominal tracheal gills, are illustrated [13] (Figures 19–24, 61, 62, 72, 78, and 91).

**Remarks**. With the exception of the different dorsal colourations (pale yellowish brown in *A. leii*, and the yellow pronotum and black elytra, which are narrowly pale-margined in *A. ficta*), we did not find any obvious features that might further distinguish these two species. We examined male genitalia, the female reproductive system (especially the shape of the MOP), and larvae. *A. leii* was first distinguished from a population from Wuhan City, Hubei Province, in China because of its pale dorsal colouration. The similarities to *A. ficta* were not mentioned, but difficulties in the definitive identification of *A. ficta* were. This is further addressed in the Discussion.


***Aquatica qingshen* Fu et al. Ballantyne sp. nov.**


urn:lsid:zoobank.org:act:658133F2-BC04-445D-905A-28E58052942D

**Figure 5 insects-15-00031-f005:**
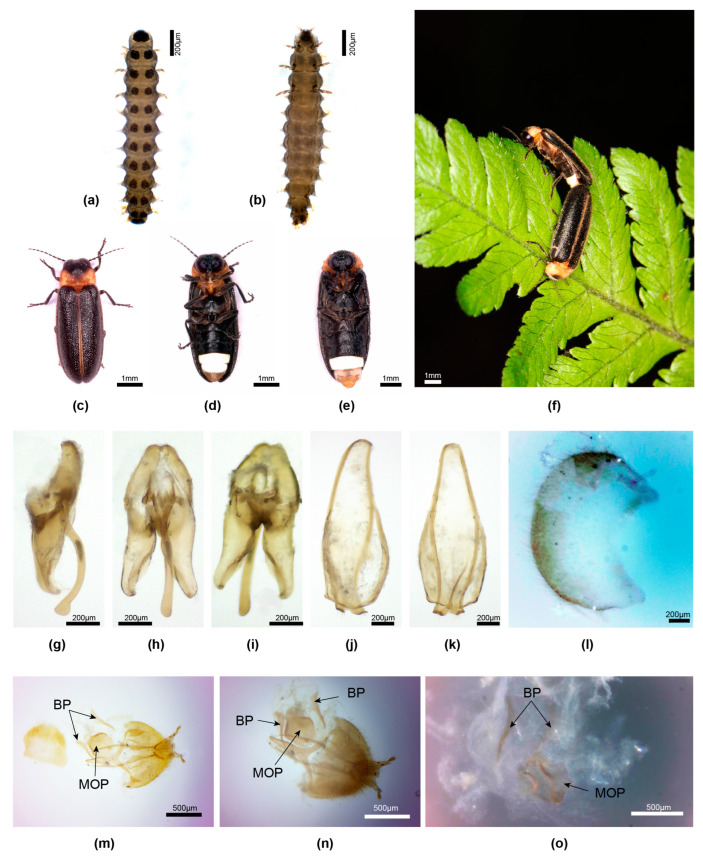
*Aquatica qingshen* sp nov. First-instar larva (**a**,**b**), male (**c**,**d**,**g**–**l**), mating pair (**f**), female (**e**,**m**–**o**). (**a**,**c**,**f**) Dorsal habitus; (**b**,**d**,**e**), Ventral habitus. (**g**–**i**) Aedeagus: (**g**) right lateral; (**h**) ventral; (**i**) dorsal. (**j**,**k**) Aedeagal sheath: (**j**) dorsal; (**k**) ventral. (**l**) Tergite 8. (**m**–**o**) Female reproductive system: (**m**) ovipositor, tergite 8, ventrite 8, MOP, and BP from below; (**n**) as for (**m**), from above; (**o**) detail MOP and BP. Panel l with anterior end to right of page; (**m**,**n**) with anterior end to left of page. Legend: BP, bursa plate; MOP, median oviduct plate.

**Types**. Holotype. Male. **CHINA**. Collector is Fu, X. H. Sichuan Province, Meishan City, Qingshen County, 8.vi. 2021. (NHMHAU). Paratypes: 4 males, 2 females, same data as for holotype; 5 males, 2 females, Sichuan Province, Le’shan City, Mt. Emei, 18.vi. 2021. (NHMHAU).

**Diagnosis**. Male: belongs to the group of *Aquatica* with darker-coloured median markings on the pronotum; one of two species with the apex of the aedeagal sheath sternite split into two truncated halves, basal piece with wide halves, LL strongly divergent along their dorsal margins, and the ML strongly curved, apically bulbous; distinguished most obviously from *A. xianning* sp. nov. by its much smaller size. Female: coloured like male, except for white LO in V6 only, clear yellow V7, 8 except for aggregation of fat body visible beneath cuticle in anterolateral portions of V7; one of two species with short ovoid MOP, distinguished from *A. xianning* female by its outline. Larvae are only known from the first stage bred from eggs, with a single protergal plate and paired separated tergal plates on segments 2–12. 

**Description of Male.** 5.3–6.3 mm long. Colour: (Figure 5c,d) PN orange-yellow, wide triangular anterior black area includes an area of retraction of fat body along anterior margin, and posterior light brown area extending laterally across 2/3 of width and almost reaching to posterior margin; anterior margin beside median marking and around neck margins narrowly black-rimmed; MS, mesonotal plates light brown; elytra black with suture narrowly pale brown; head, antennae, and palpi black; pro- and mesosterna deep orange, venter of metathorax black; legs black, except for coxae 1 and 2, which are slightly paler orange on ventral surface; abdomen black, except for white LO in V6 and anterior half of V7, posterior half of V7 semitransparent, clear (no underlying fat body); T8 pale semitransparent yellow; T7 semitransparent, clear in posterior 1/3, black in anterior 2/3. 

**Male genitalia***. Aedeagal sheath*: (Figure 5j,k) sheath sternite L/W 2.7, base wide, broadly rounded, apex split into two equal-sized apically truncated pieces, with angulated corners approx. 90°, a single pointed projection along each margin in posterior area, and posterior half of sternite broad; R and L tergite arms join side of sternite approximately equidistant from sheath base. *Aedeagus:* (Figure 5g–i) L/W 2.9; basal piece with wide lateral halves, anterior margin narrowly dark-coloured, well sclerotised, median anterior area not expanded, not truncated; LL divergent along most of their length, widely divergent along inner dorsal margins, with narrow inturned apices that appear hook-like; ML slender, wider in basal 1/3, strongly curved when viewed from the side, with apex rounded and expanding in dorso-ventral plane only.

**Female**. 5.7–7.1 mm long. Colour: coloured like male, except for V6 with white LO, V7, 8 pale orange, semitransparent, with fat bodies clustered across anterior area of V7 (Figure 5e). *Reproductive system*: (Figure 5m–o) bursa plates elongated, wider at base, with elongated narrowed acute tips; MOP ovoid, with median anterior margin slightly projecting and rounded, sides rounded, posterior margin projecting, truncated; paired, slightly curved, elongated ridges in lateral areas; not possible to determine the extent of MOP relative to both the median oviduct and vagina.

**Larva**. (Figure 5a,b) Fu was able to breed larvae to the first stage only, which confirmed their aquatic status; with a single plate on protergum, paired separated tergal plates on body segments 2–12, and lateral abdominal gills on abdominal segments 1–9.

**Etymology**. This species is named after the location where adults were first discovered and collected. Qingshen is here treated as a noun in apposition to retain the original Chinese nomenclature.


***Aquatica wuhana* Fu et al. Ballantyne 2006**


**Figure 6 insects-15-00031-f006:**
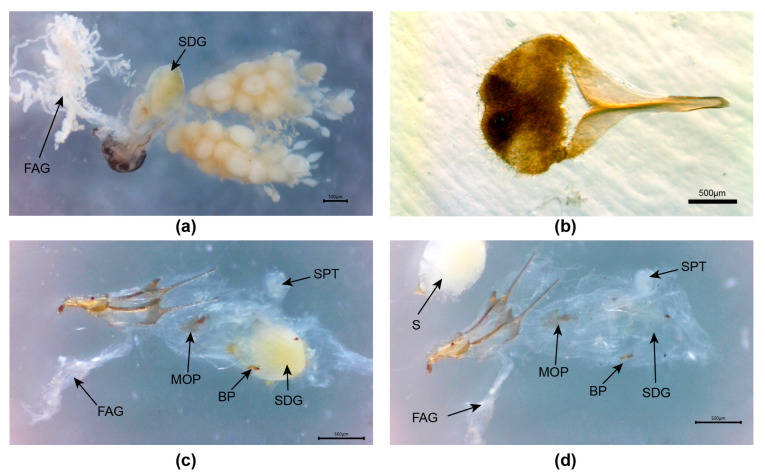
*Aquatica wuhana.* Female. (**a**) Intact reproductive system from above; (**b**) ventrite 8; (**c**,**d**) dissected reproductive system with ovipositor to left of picture. Panel (**a**) has anterior end to left of page; (**b**–**d**) anterior end to right of page. Legend: BP, bursa plate; FAG, female accessory gland; MOP, median oviduct plate; SDG, spermatophore-digesting gland; SPT, spermatheca; S, spermatophore remnants.

**Specimens examined**. **CHINA**. Collector is Fu, X. H. Hubei Province, Wuhan City, Jiang Xia District, E’Jia’Bian Village, 20 males, 6 females, 28. v. 2008; Hubei Province, Huangshi City, Yangxin County, 3 male, 1 female, 29. v. 2018; Hubei Province, Xianning City, Tongshan County, 6 males, 2 females, 2. vi. 2022; Hubei Province, Ezhou City, Liangzihu District, 5 males, 2 females, 26. v. 2023. (NHMHAU).

**Diagnosis**. Male. One of a group of four *Aquatica* species with dark-coloured median markings on the pronotum; distinguished from *A. xianning* sp. nov. and *A. qingshen* sp. nov. by the non-curved ML, non-divergent LL, and the entire apex of the aedeagal sheath sternite (*A. xianning* sp. nov. and *A. qingshen* sp. nov. have strongly curved ML and strongly divergent LL, and the sheath sternite apex splits into two divergent pieces); distinguished from *A. lateralis* by the restricted brown pronotal markings and the non-toothed inner dorsal margins of LL (*A. lateralis* has extensive dark pronotal markings extending from anterior to posterior margins and toothed inner dorsal margins of the LL). Female MOP elongated, longer than wide, posterior area with margins converging to a broadly rounded apex, median anterior margin narrowly produced, sides expanding. 

**Male genitalia***. Aedeagal sheath*: sheath sternite L/W 4.0, base wide, broadly rounded, apex entire, rounded, a single pointed projection along each margin in posterior area, posterior half of sternite broad; R and L tergite arms join side of sternite approximately equidistant from sheath base. *Aedeagus:* L/W 3.0; basal piece with wide lateral halves, anterior margin narrowly dark, well sclerotised, median anterior area bearing paired fine rounded projections beside median line; LL closely approach in basal 1/3 of their dorsal length, then diverge, and are slightly obliquely truncated along outer edges; ML slender, wider in basal 1/2, not strongly curved when viewed from the side, with apex rounded and expanding in dorso-ventral plane only.

**Female**. 7.1–8.9 mm long. Colour: coloured like male except, for white LO in V6 only, V7, 8 pale orange, semitransparent, with fat bodies clustered across anterior area of V7. *Reproductive system*: (Figure 6) bursa plates elongated, wider at base with elongated narrowed acute tips; MOP elongated, longer than wide, posterior area with margins converging to a broadly rounded apex, median anterior margin narrowly produced; not possible to determine extent of MOP relative to both the median oviduct and vagina.


***Aquatica xianning* Fu et al. Ballantyne sp. nov.**


rn:lsid:zoobank.org:act:0F7865C8-0F5B-4E49-B938-7597B2434D2B

**Figure 7 insects-15-00031-f007:**
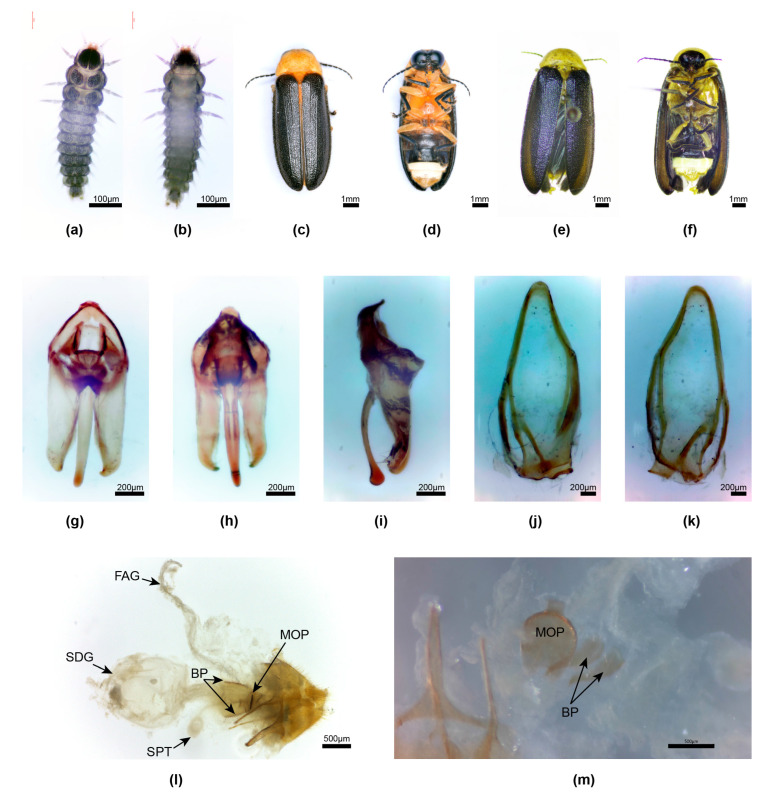
*Aquatica xianning* sp. nov**.** First-instar larva (**a**,**b**), male (**c**,**d**,**g**–**k**), female (**e**,**f**,**l**,**m**). (**a**,**c**,**e**) Dorsal habitus; (**b**,**d**,**f**) Ventral habitus. (**g**–**i**) Aedeagus: (**g**) dorsal; (**h**) ventral; (**i**) left lateral. (**j**,**k**) Aedeagal sheath: (**j**) ventral; (**k**) dorsal. (**l**,**m**) Female reproductive system: (**l**) intact system with SDG to left; (**m**) detail MOP and BP from below (anterior margin of tergite 8 to bottom left). Panel (**l**) with anterior end to left of page. Legend: BP, bursa plate; FAG, female accessory gland; MOP, median oviduct plate; SPT, spermatheca; SDG, spermatophore-digesting gland.

**Types**. Holotype. Male. CHINA. Collector is Fu. X. H. Hubei Province, Xianning City, White Spring Village, 28.v. 2021. (NHMHAU). Paratypes: 4 males, 2 females, same data as for holotype (NHMHAU).

**Diagnosis**. Male: pronotum yellow, with very diffuse, pale, anteromedian brown markings visible only in some specimens; retraction of fat bodies beneath cuticle may confuse interpretation of colour, especially along anterior margin, where the dark head is visible beneath transparent cuticle; elytra very dark brown, almost black without paler margins; one of two species with the tip of the sheath sternite split into two truncated halves, LL strongly divergent along dorsal margins, and ML strongly curved, apically bulbous; distinguished most obviously from *A. qingshen* sp. nov. by its much larger size (8–12 mm vs. 5.3–6.3 mm) and wider basal piece, which has the anterior margin sclerotised and the mid-anterior margin thickened. Female: coloured like male, except for abdominal V7, 8 semitransparent; MOP short, ovoid, not extending into both median oviduct and vagina, median anterior projection shortened, apically truncated, posterior area expanded, sides rounded, posterior margin emarginated. 

**Male**. 8–12 mm long. *Colour*: pronotum yellow, irregular retraction of fat bodies beneath cuticle makes colour interpretation difficult; small pale diffuse brown anteromedian markings not visible in two males; clear semitransparent area along anterior margin arising from retraction of fat bodies, with underlying black head visible along this margin; fat bodies in median and posterior areas irregularly retracted; mesonotal plates yellow, MS brownish yellow; elytra black; head, antennae, and palpi black; ventral surface of pro- and mesothorax pale brownish orange, that of metathorax brown; legs 1 and 2 black, except for pale brown coxae and trochanters, and a brownish orange narrow base of femora; legs 3 like legs 1 and 2, except for brownish orange median apex of coxae; ventral abdomen all black, except for white LO in V6, and white LO in V7 retracted to anterior half with posterior area semitransparent, without underlying fat body; dorsally reflexed margins of V2, 3 mid-brown; T 4–6 and dorsally reflexed margins of V4, 5 very dark brown; T7 with brown markings in anterior half having an irregular posterior margin, posterior area of T7 cream; T8 pale yellowish semitransparent; dorsally reflexed margins of V6, 7 white.

*Aedeagal sheath*: (Figure 7j,k) sheath sternite L/W 3/1, base wide and broadly rounded, apex split into two equal-sized apically truncated pieces, with angulated corners approx. 90°, a single pointed projection along each margin in posterior area, posterior half of sternite broad; L tergite arm joins side of sternite anterior to that of R tergite arm. *Aedeagus:* (Figure 7g,h) L/W 2.4; basal piece incompletely subdivided; anterior margin narrowly dark-coloured, well sclerotised, median anterior area slightly expanded, truncated; LL parallel-sided along most of their length, widely divergent along inner dorsal margins, with narrow inturned apices; ML slender, slightly wider in basal 1/5, strongly curved when viewed from the side, with apex rounded and expanding in dorso-ventral plane only. 

**Female**. 10–13 mm long. Colour: (Figure 7e,f) like male, except for very diffuse darker markings on pronotum, extent unclear, V7, 8 semitransparent, yellowish with fat bodies clustered in anterior areas of V7. *Reproductive system*: (Figure 7l,m) V8 with darker anterior apodeme separated by semitransparent area from remainder of brown ventrite; bursa plates elongated, wider at base with narrowed acute tips; MOP ovoid, with shortened, apically truncated, median anterior projection, posterior area expanded, sides rounded, posterior margin emarginated; not possible to determine extent of MOP relative to both the median oviduct and vagina.

**Larva**. (Figure 7a,b) Bred only to first instar to confirm aquatic status; large plate on protergum slightly separated in posterior area; remaining two thoracic and eight abdominal terga with paired tergal plates widely separated in median line; tergum 9 with single tergal plate.

**Etymology**. This species is named after the location where adults were discovered and aquatic larvae were confirmed. Xianning is here treated as a noun in apposition to retain the original Chinese language and nomenclature.

**Remarks**. Adults were observed flashing in trees or bamboos along streams in a valley. It was not possible to breed the larvae beyond the first instar.

### 3.7. Characteristics of the Mitogenome of Two New Species of Aquatic Fireflies

The complete mitogenome of the two new species of firefly is a typical circular molecule with a total length ranging from 16,249 bp (*Aquatica qingshen*) to 16,350 bp (*Aquatica xianning*) (Appendix A). Like most Coleopteran insects, there are 31 coding genes, including 13 PCGs (protein-coding genes); 2 RNAs; 22 tRNAs; and 1 major CR (control region). Among 37 genes, *nad5*, *nad4*, *nad4L*, *nad1*, 8 tRNAs (tRNA-Gln, Cys, Tyr, Phe, His, Pro, Leu (CUN), and Val), and 2 RNAs were located on the reverse strand; *nd2*, *cox1*, *cox2*, *cox3*, *atp6*, atp*8*, *nad3*, *nad6*, *cytb,* and 14 *tRNA* were located on the direct strand (Appendix A). 

### 3.8. Phylogeny

The monophyly of the genus *Aquatica* is strongly supported in six trees (BS = 100, PP = 1.0) (Figure 8). The position of *Aquatica* as sister to the genus *Nipponoluciola* is stable across BI and ML, consistent with previous studies based on morphological characters [4]. *A. qingshen* and *A. xianning* are closely related and classified into a single clade, which is strongly supported in six trees (BS = 100, PP = 1.0). The Chinese population of *A. lateralis* is closely related to the Japanese population, with a p-distance of 0.0947 (Appendix A), although they cannot be separated using morphological characters. Although the colour pattern differs between *A. leii* and *A. ficta*, their genetic distance is very close (p-distance of 0.0036) (Appendix A). Issues of a taxonomic and nomenclatural nature for *ficta* that will prevent the easy resolution of this problem (regarding whether they are the same species) are given in the Discussion. Large integrated datasets are required to improve phylogenetic resolution between Lampyridae taxa in the future, such as greater integrated datasets of nuclear genes, mitogenomes, and morphological characters.

The interrelationships among species of *Aquatica* varied depending on the molecular markers and the phylogenetic technique used. The most significant disagreement was in the placement of the *A. lateralis* clade (Figure 8). The phylogenetic position of *A. lateralis* as sister to (*A. ficta* + *A. leii*) + *A. wuhana* is supported by the PCGs (85/1) and PCGsRNA (89/1) trees. Although the branching pattern of *Luciola curtithorax* and *Sclerotia substriata* inferred from the AA and PCGs was inconsistent, the conflicting topologies were not statistically supported. 

## 4. Discussion

The aquatic genus *Aquatica* is now known to comprise seven species, with two of them being new. Here, a combination of an analysis of mitogenomes, newly determined for the two new species, with the morphological features of three stages (male, female, and larva), allowed us to define the genus from both aspects in an integrative taxonomy approach at the generic level in the Luciolinae. The monophyly of the genus was established from molecular information. Species are now known and described, re-described, and keyed from males, females, and larvae with the following exceptions: *A. hydrophila* female was not dissected, and a larva of the Chinese population of *A. lateralis* has not yet been found. Information concerning all species now assigned to *Aquatica* is consolidated here (Appendix A).

Female reproductive anatomy (except for that of *A. hydrophila* female) is expanded with details of the unique structure of the median oviduct plate in several species. It is possible that only the elongated MOP seen in *A. leii* and *A. ficta* extends from the median oviduct into the vagina. A female accessory gland, which may provide material for the eggs (nutrition or eggshell components), is identified in all but *A. hydrophila* (Appendix A). The similarity of this gland to strands of fat body surrounding the reproductive system may have allowed its presence to have been inadvertently overlooked in previous examinations. 

The definitions of the aquatic status for the Luciolinae are slightly amended for ease of usage. However, we were unable to differentiate generic features for the two genera with bottom-feeding, gilled larvae.

While we applaud the philosophy of [12], we simply do not know enough of Luciolinae morphology, genetics, behaviour, and ecology to be able to integrate all the features that he mentions. Only this and two other papers [34,46] have attempted an integrative approach to the genera that they addressed. Very few taxonomists report the kind of criteria or evidence that are critical to their species concept, but even that is compounded by the lack of appreciation of the pursuit of taxonomy and the very few firefly taxonomists. 

Species are distinguished here by a combination of morphological features of all life stages, in combination with genetic evidence. Most specimens in this study are from individual isolated populations only discovered by the extensive collecting activity of Fu, who was in turn guided in this collection by the recognition (at night) of the male flashing patterns, which are yet to be studied in this genus.

The correct identification of species inhabiting bodies of water is essential for measures that address their conservation. Our molecular analyses suggest that *A. ficta* and *A. leii* may well be the same species, as do [47,48,49,50], which were, however, all based on the same genome for each species. However, assessing this possibility is currently compounded by taxonomic and nomenclatural issues, and it is possible that, in its original description, *Luciola ficta* may not have been correctly identified. Given their morphological similarities (in aedeagal and aedeagal sheath structure), *A. leii* and *A. ficta* were initially differentiated [1] on the basis of colouration (*A*. *leii* with very pale yellowish-brown elytra and one black abdominal ventrite; *A*. *ficta* with almost black elytra narrowly pale-margined and black ventrites anterior to the light organs). However, a Hong Kong population, with an apparently intermediate colouration of mid-brown elytra and black abdominal ventrites anterior to the light organs, was interpreted as *A. leii* [51], with *A*. *ficta* considered absent from Hong Kong.

None of the references to *A. ficta* subsequent to the original description [1,3,4,16,36,37] is consistent with the description of a male abdomen with three white terminal ventrites (but all do appear to refer to the same species). We do not know of any Luciolinae with a three-segmented light organ, but misinterpretations can occur if ventrite 5 is pale-coloured [3].

Did the original author simply misinterpret the situation [39]? The light organ material in ventrite 7 is withdrawn into the anterior half of that ventrite and might have been interpreted as an additional segment. No male types remain to permit confirmation. Or did he actually describe three females?

The identification of a Taiwanese population as *ficta* [52], without any commentary on morphology or precise location, led to a wide characterisation of many populations in both mainland and offshore China as this species, albeit all with males having only two pale terminal segments, all apparently having aquatic larvae with abdominal gills, and being from a similar habitat. 

If only morphology is addressed, then colour patterns alone cannot define *A. ficta* (it has to be dissected). However, several Luciolinae species have already been described with similar patterns to those of *ficta* and must be considered in this argument. The following all have dark brown to black elytra with pale margins (extent not specified, however): *Luciola binhana* Pic from Vietnam; *L. robusticeps* Pic from the East Indies; and *L. fukiensis* Pic, *L. kalapperichi* Pic, and *L. limbalis* Fairmaire (female only) from China. None of these species has a type associated, there is no information about larval habitat, and all were treated as *incertae sedis* [3]. 

*A. wuhuna* has a similar male genitalic pattern to that of *A. leii* and *A. ficta*, but it differs most obviously in the dark-coloured median marking on the pronotum, the almost completely black elytra, and the distinctive shape of the MOP (shorter ovoid rather than elongated subrectangular and parallel-sided). It may be that future studies incorporating more females may help differentiate these species.

An attempt to stabilise the nomenclature by nominating a female lectotype [16] only added to the problem. Luciolinae taxonomy is heavily male-based, and we are not always able to identify females not taken with a male to genus, let alone to species [3]. Only in certain genera like *Aquatica* could features of the female reproductive system possibly provide information that might be used for both generic and species identification [3]. The lectotype was not dissected. However, the ICZN rule 74.1 indicates that a lectotype must be designated from the original type series (of which only three males were listed), indicating that this designation is invalid [40].

The type specimen of *Luciola lateralis* has not been investigated using any taxonomic approach [1,4,16]. The populations in Korea and Japan have been shown to be genetically different [6], and the possibility of yet another species in China will be investigated once the aquatic nature of the larvae is confirmed. In contrast, the Chinese population of *A. lateralis* is closely related to the Japanese population but with a far greater genetic distance, although they cannot be separated using morphological characters.

## Figures and Tables

**Figure 8 insects-15-00031-f008:**
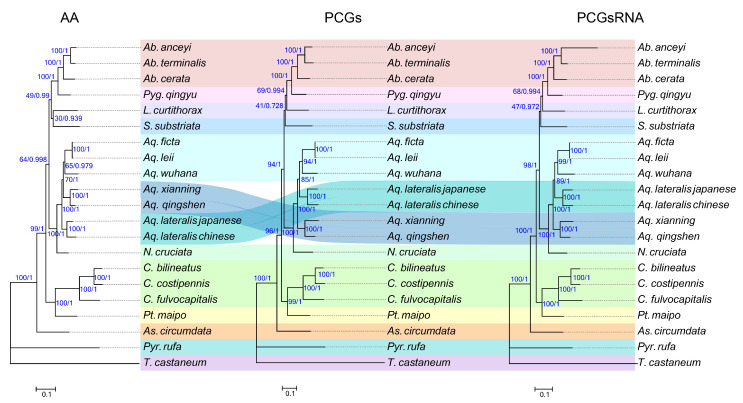
Phylogenetic tree produced for AA dataset (left), PCGs (middle), and PCGsRNA (right). Support at nodes (from left to right) are bootstrap support values obtained via maximum likelihood and posterior probability obtained via Bayesian inference analyses. AA 13 amino acid sequences of all PCGs; PCG 13 nucleotide sequences of PCGsRNA; PCGsRNA PCG plus 22 tRNA and 2 rRNA.

## Data Availability

Data are contained within the article or Appendix A.

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
