# Peer review of "An Overview of Aquatica Fu et al., a Phylogeny of Aquatic Fireflies Using Mitochondrial Genomes, a Description of Two New Species, and a New Record of Aquatic Fireflies in China (Coleoptera: Lampyridae: Luciolinae)â€"

_insects, 2024, doi:10.3390/insects15010031_

Round 1
Reviewer 1 Report
Comments and Suggestions for Authors
Insects-2708040: An Overview of Aquatica Fu et al., a Phylogeny of Aquatic Fireflies using Mitochondrial Genomes, Description of two New Species, and a new Record of Aquatic Fireflies in China (Coleoptera: Lampyridae: Luciolinae).
This manuscript lists four aims:
1. to contribute to the knowledge of Aquatica by providing descriptions of two new species, and one new record.
2. to update and consolidate the morphological, taxonomic and historical information available for the genus and reassess the varying definitions of aquatic status.
3. include a novel examination of aspects of the female reproductive system.
4. phylogenetic analyses of mitogenomes to determine the generic phylogenetic position and the relationships among its constituent species, as well as to assess its monophyly.
The authors did an excellent job describing the morphological traits of Aquatica species and integrating two new species into a dichotomous taxonomic key, alongside previously described Aquatica species. Especially the addition of female morphology and the association of males, females and larvae is a major contribution to the field. The overview of aquatic categories among Luciolinae larvae is also an asset of this study.
In the context of their fourth aim the authors show that all Aquatica taxa in their analysis form a clade, but several clarifications are needed in this section, as well as an improved integration of the molecular and morphological findings towards a more in-depth discussion of the species status of several Aquatica taxa based on current evidence. Presently the discussion is undeveloped and could be improved greatly.
Selected Clarifications Needed:
Line 31: “A Chinese population of A. lateralis was found to be morphologically similar to the Japanese population but genetic distances show it to be a distinctive species.”
Note: There is no specific evidence provided in this manuscript for this statement. Based on which criteria are these different species? There is no species delineation conducted in this study, only a single specimen (or mitogenome) is used from each population across the Aquatica species. How do the authors distinguish between variation by geographic distance within a species from distinct species? What is the threshold and how can it be justified? Ahrens 2023 (cited by the authors) cites a whole body of literature on that, esp. multispecies coalescent, geography, role of evolutionary process etc. - Or is it solely based on comparable distances (branchlengths) between A. xianning and A. quingshen (or Curtos b. and c) (Fig. 8) – along with independent clear evidence for being different species? What other evidence (including morphology) supports this? Please clarify your specific criteria, provide the data (results) and explain. Please integrate all evidence to support the conclusion in the discussion.
Also Line 46: The authors cite and endorse Ahrens with respect to “morphology as a mandatory diagnostic trait”, however they seem to ignore it here (two populations are “similar”). What are the individual male and female morphological traits in these two populations that may suggest that they could be reproductively isolated and would support a two species hypothesis - or vice versa: supporting one species? Which traits differ (what is “similar”)?
124: “For genome sequencing, Fu collected additional dozens of individuals of each species immediately preserved in anhydrous ethanol, followed by preservation at -40°C in the laboratory prior to DNA extraction.”
Note: Were several individuals pooled for genome sequencing? How many? From the same location? Or do genomes represent a single specimen? Please clarify (and also add this to discussion, if pooled).
130: “…an excellent integrity of DNA molecules were observed”
Note: Please clarify: how was the “integrity” determined (criteria)?
163: “All aligned genes were concatenated into three datasets”
Note: Please clarify the aim of each data set here, e.g. AA versus nucleotides (and pick that back up in discussion, along with discussion why taxa may have flipped between AA and nucleotide data sets: How common is that in mitogenome analysis? Implications?). The context is missing in both methods and results/discussion.
190: “Both morphological and molecular phylogeny confirm their distinct species status …”
Note: This statement is contradicted by the phylogeny in Fig. 8: neither marker set shows a difference between A. leii and A. ficta (same branch lengths). Please correct. Could they be the same species? Discuss this in depth with specific lines of evidence and literature in the discussion: expand on 819).
774: The monophyly of genus Aquatica is strongly supported in six trees (BS = 100, PP = 1.0) (Figure 8).
Note: In Figure 8 only three trees are shown: are these the ML trees with both bootstrap and pp values? Are there any differences between best ML and majority BI trees? Please clarify.
781: “Large integrated datasets are required to improve phylogenetic resolution between Lampyridae taxa in the future, such as greater integrated datasets of nuclear genes, mitogenomes and morphological characters.”
Note: This is a very general and vague statement (please move this section to discussion and clarify). Please elaborate and explain (in the context of Aquatica) in more detail - explain what mitochondrial versus nuclear data sets versus morphology contribute and which specific questions can be addressed by each/by integrating all of these. This would also be a good opportunity to discuss which data would be needed in future studies to test the species hypotheses for A.lateralis, A. ficta and A. leii developed in this manuscript.
786: Figure 8. Phylogenetic tree produced for AA dataset (left), PCGs (middle) and PCGsRNA (right)
Note: please add what abbreviations stand for to the figure legend; ML trees shown?
Discussion section
Note: The discussion section is the weak point of this manuscript. It should integrate the findings of this study with questions/issues raised and generate new and specific insights (supported or contradicted by the literature). Right now, different topics are simply listed and findings restated, but not discussed or integrated. The manuscript would benefit from laying out the arguments clearly and being as specific as possible (specific species, traits, how evidence connects, etc.). One big element in the introduction is the explicitly stated need for integration of molecular and (mandatory) morphological evidence for the species concept (Ahrens) - this element is currently missing in the discussion and the manuscript would greatly benefit from a thoughtful integration of specific results, and an in-depth discussion what this implies for species hypotheses within Aquatica (and how this will inform future studies).
Examples: specifics needed
795: “With two exceptions all species are now known”
Note: Please list the two exceptions/species here (don’t rely on the reader to figure it out).
801: “A female accessory gland, which may provide material for the eggs (nutrition or egg shell components) is newly identified in several species.”
Note: Please specify: which are new? The two new species in this study? Also please add a summary, e.g. identified in 5 of the 6 species studied to date? and identify where this can be found (Table S2). This information should be readily available/summarized for the reader, and not rely on the reader to flip through the manuscript and figures and tables to find.
Examples: clear (and applied) message needed
808: “Determinations of the complete mitogenome are almost always accompanied by phylogenetic trees notwithstanding the number of species addressed. If too few species are included, then wider interpretations are difficult. It is possible however, that results may mislead, and give rise to the one thing we wish to avoid, taxonomic instability [41, 811 42].”
Note: What is the message (and discussion) here? How does this apply to the present study and inform the next steps? Are all important species included here? What is missing? How can the results here potentially be misleading? For example, how does this apply to the species questions raised? What should be included in a follow-up study and why? Please clarify and be as specific as possible.
813: “Where the nomenclature of species included does not correspond to the most recent taxonomy, then the genus Luciola will be perceived as paraphyletic [43, 4]. This occurs especially where the species lateralis Motschulsky is attributed to Luciola rather than Aquatica [4]. The genus Aquatica however was not erected until 2010 [1], and we list all species presently assigned to Aquatica including any that previously stood under the genus Luciola.”
Note: What is the context here? Is that methods? Or what is the purpose of this section in the discussion? Where do you “list” the taxa - in a table? Are they all labelled as Aquatica in the phylogeny? Or where could confusion occur? Which specific point are you addressing here?
Examples: Integration (of specific results and literature) needed
819: “Molecular analyses [8-10, 44] suggest that A. ficta and A. leii may well be the same species. Our mitogenome phylogeny also showed genetic distance between A. leii and A.ficta is very close, although color patterns differ. Because the species is based on a female lectotype designation [1] it was considered any identifications of males in this species would need further study, before the apparent close relationship between the two can be further addressed.
Note: Please transform this section into arguments for and/or against one or two species. Right now, this is a list of facts without discussion or argument one way or another.
(1) “Our mitogenome phylogeny also showed genetic distance between A. leii and A. ficta is very close, although color patterns differ.”
Note: What does "close" mean? Can you quantify? Based on the branch lengths shown in Fig. 8 A. leii and A. ficta look identical for all 3 data sets. Since different color patterns are noted here: specify how they differ; it would be a good opportunity here to discuss color patterns as potential diagnostic features (e.g. how labile/ecological versus stable/taxonomic traits are these in the literature? How much weight should these get in favor or against one or two species?). In addition, in the spirit of integrating both molecular data and morphology, please discuss: what are the morphological traits that would support or contradict the hypothesis that these are the same/different species?
(2) Connect all lines of evidence into an argument for or against two species (based on current evidence including this manuscript). Example argument: your mitochondrial data (specificy) suggest that these are the same species, this is supported by other molecular studies (specify: are these all mitochondrial or also nuclear evidence? REF). Morphology: which traits and how do they contribute in favor or against the same species?. Then connect "female type" (what evidence is missing: which male data needed? and why? What is your conclusion based on this combined evidence? Be be as specific as possible so the reader can follow a clear argument.
(3) Also, what does “further study” and “further addressed” mean? How would one do it? Which traits need to be analyzed and what do you expect to find in each case to be able to decide whether it is one or different species?
Overall, the discussion would be greatly improved if specific evidence is used to make clear points and develop convincing arguments for one or two species. This also applies to taxon sampling and the contribution of different data sets.
Comments on the Quality of English LanguageThere are a few extra words in some sentences, but overall this manuscript is well written.
Reviewer 2 Report
Comments and Suggestions for Authors
This is a very nice revision of the genus Aquatica with the addition of two new species. The inclusion of keys for males, females and larvae is especially appreciated.
I would avoid the use of the words feature(s) and breeding, and instead use character(s) and rearing.
Otherwise well done.

Comments on the Quality of English LanguageJust a few minor problems.
Author Response
Thank you for all your work on this. We have accepted all your suggestions which are indicated in colour in the attached draft (except for all the conversions to an en dash).

Reviewer 3 Report
Comments and Suggestions for Authors
This is a valuable and rigorous taxonomy and systematics work on aquatic fireflies, with combining molecular information and morphological taxonomy. Two new species of genus Aquatica described from features of males, females and larvae, and the relationships of these species with other aquatic fireflies is established. The research expanded definitions of aquatic status for Luciolinae fireflies. I have no objection to the writing and conclusion of the article. But it would be perfect if the resolution of the genitalia photographs could be improved.
Round 2
Reviewer 1 Report
Comments and Suggestions for Authors
The authors addressed all areas of concern except one: there is still confusion about "species delineation" and I suspect this is due to the fact that the authors (1) would like to distinguish species, and (2) used the "species delineation" website to calculate pairwise genetic distances. However, no actual "species delineation" in the strict sense (with within populations sampling for each species to generate molecular bar code gaps between larger samples) was conducted here.
Suggestion: please remove the “species delineation” terminology; this is misleading - and unnecessary. It seems what was done here was using the species delineation website to calculate the pairwise genetic distances between all established and tentative species in this study (one specimen each, with 2 individual geographic samples for 1 species). The p-values can be discussed without claiming “species delineation”. The authors used genetic distances and species trees from the 3 different/overlapping data sets to evaluate how distinct the species in their study are from each other. This is a perfectly good approach and should not be confused with "species delineation" as a specific method for a different purpose.
Here are my suggestions for the text to avoid this problem (edits in blue):
The pairwise genetic distances (p-distance) between all taxa in this study were calculated using ABGD web (https://bioinfo.mnhn.fr/abi/public/abgd/abgdweb.html) for species delimitation with the Kimura two-parameter (K80) distance model.
All aligned genes were concatenated into three datasets using PhyloSuite v1. 2. 2.: (1) 169 combine the 13 amino acid sequences of all PCGs (AA); (2) combine the 13 nucleotide 170 sequences of all protein-coding genes (PCGs); (3) combine the PCGs, 22 tRNA and two 171 rRNA genes (PCGsRNA). ModelFinder [29] was used to select the best-fit model follow-172 ing BIC criterion in the three generated datasets for IQ-TREE and MrBayes, respectively. 173 The best-fit model for AA is mtMet+F+R4 and mtREV+F+I+G4, model for PCGs and PCGs-174 RNA are GTR+F+R4 and GTR+F+I+G4. This diverse dataset was used here for the first 175 time to allow us to explore the effectiveness or otherwise of these molecular markers in 176 species delimitation.
Note: model testing is used to find the best model for the phylogenetic reconstruction (given the loci used), not to evaluate "effectiveness of molecular markers". What is compared here are the three resulting phylogenies and how consistently they support the species relationships.
2. Lines 861-867 Species are delineated are distinguished here by a combination of morphological features of all life stages, allied with genetic evidence. Most specimens in this study are from individual isolated populations only discovered by the extensive collecting activity of Fu, who was in turn guided in this collection by the recognition (at night) of the male flashing patterns, which are yet to be studied in this genus. P-distances were used to further confirm identification of two similar species (Table S2), but given that this type of investigation is still in its
infancy, the parameters used will eventually need to be evaluated from a much wider selection of species. A ‘justified threshold for species boundaries has yet to be determined’ [46].
And just a minor clarification:
Rephrase for context:
However assessing this possibility is presently compounded by taxonomic and nomenclatural issues. Luciola ficta may not be correctly identified.
Suggestion:
However, assessing this possibility is presently compounded by taxonomic and nomenclatural issues, and it is possible, that in its original description Luciola ficta may not have been correctly identified.
Author Response
Thank you for your suggestions which I have accepted and they appear in the modified document attached (with apologies for what is now rather lurid highlighting but I was running out of strong easily identifiable colours). As an insect taxonomist I am most appreciative for this additional information you suggested.
